# Methodological issues in visible LED therapy dermatological research and reporting

David Robert Grimes[1,2]*

**1** TCD Biostatistics Unit, Discipline of Public Health and Primary Care, School of Medicine, Trinity College Dublin, Dublin, Ireland, **2** School of Physical Sciences, Dublin City University, Dublin, Ireland

\* davidrobert.grimes@tcd.ie

## Abstract

### Background

The advent of mass-market Light Emitting Diodes (LEDs) has seen considerable interest in potential dermatological applications of LED light photobiomodulation (PBM) for a range of conditions, with a thriving market for direct-to-consumer LED treatments, including red light, blue light, and yellow light wavelengths. Evidence of efficacy is however mixed, and studies report a wide range of irradiances and wavelengths as well as outcome measures, rendering interpretation, comparison, and even efficacy evaluation prohibitive and impeding evidence synthesis.

### Methods

This work establishes a model for comparing patient received doses, applying this to existent studies to ascertain potential inhomogeneity in reported doses and wavelengths employed. Patient doses were contrasted to equivalent solar exposure time needed to achieve fluences reported at specified wavelengths in the red light (RL), blue light (BL), and yellow light (YL) portion of the spectrum, yielding a comparison of reported doses to typical solar irradiance at the Earth's surface. Methodological aspects including dose validation, blinding, and bias were also analysed.

### Results

27 relevant studies for dermatological conditions including acne vulgaris ($n = 9$, 33.3%), wrinkle-reduction ($n = 5, 18.5\%$), wound-healing ($n = 3$, 11.1%), psoriasis severity ($n = 3$, 11.1%), and erythemal index ($n = 7$, 25.9%) were assessed. Outcome measures were highly heterogeneous between studies, with total patients ranging from $14 - 105$ (median: 26). Fluences and wavelengths used in treatment differed over three orders of magnitude across studies even for the same conditions ($0.1\ J\ cm^{-2} - 126\ J\ cm^{-2}$, median: $40.5\ J\ cm^{-2}$). Derived equivalent solar time ranged from 0.01-19.35 hours (median: 3.3 hours), with central wavelengths between 405nm

**Data availability statement:** Code to run all the analysis in this work is hosted on GitHub (https://github.com/drg85/LEDcheck) and further analytical software in various languages is available by request from the author.

**Funding:** This work has been funded by the Wellcome Trust (Grant number 214461/A/18/Z). The funders had no role in study design, data collection and analysis, decision to publish, or preparation of the manuscript.

**Competing interests:** DRG declares no support from any organisation for the submitted work; no financial relationships with any organisations that might have an interest in the submitted work in the previous three years; no other relationships or activities that could appear to have influenced the submitted work.

(BL) - 660nm (RL). No studies reported any dose validation, 10 (37.0%) were sponsored by the device manufacturer with a further 3 (11.1%) conducted by commercial dermatology practices offering the therapy under investigation. Assessors were unblinded to the treatment/ control groups in 33.3% ($n = 9$), while a further 9 (33.3%) did not have any non-light control group, leaving only 33.3% ($n = 9$) with both control group and blinded outcome assessment.

## Conclusions

Results of this analysis suggest that fluences, wavelengths, and effective dose vary inconsistently between studies with often scant biological justification. This analysis suggests that better dose quantification and understanding of the underlying biophysics as well as plausible biological justifications for various wavelengths and fluences are imperative if LED therapy studies for dermatology are to be informative and research replicability improved.

## Introduction

In recent years the home medical treatments have exploded in popularity, with the market estimated to be worth $56.45 billion by 2027, the majority comprised of therapeutic equipment [1]. Home dermatological treatments are especially popular with the wider public, sold with the allure of reducing expensive visits to dermatological experts for a range of conditions. The advent of cheap light-emitting diodes has resulted in a huge and growing array of LED-based skincare treatments with FDA registered devices on the market, typically in the red-light (RL), blue-light (BL), or yellow-light (YL) portion of the visible spectrum [2]. Collectively known as low intensity photobiomodulation (PBM) devices, the many products on offer and claims around them creates some confusion – many of these projects are not rated by the FDA for efficacy, only for safety and similarity to existing products, with the FDA clarifying that *"the mechanism of actions for PBM for different clinical indications is not fully understood. Outcomes are dependent on many factors such as wavelength of light, fluence, irradiance, pulsing parameters, and beam spot size"* [3]. This note of caution from the FDA has not stopped a roaring trade in PBMs online, with LED therapies trending repeatedly on social media with tens of millions of views [4].

Despite many influencers selling expensive home LED PBM devices including face masks and LED arrays to an eager audience, the evidence of efficacy for many of the claims made is markedly mixed. A 2023 review [5] of 31 studies using standardized mean differences looked at RL, BL, and YL LED therapies for several skin conditions. It reported that both RL and BL were effective for acne vulgaris, skin rejuvenation (wrinkle reduction), and psoriasis treatment, and found that all RL, BL, and YL were all effective for reducing erythema index rate, whereas there was no evidence for BL having a positive impact on wound healing. This itself is curious, given that these are ostensibly very different wavelengths at opposite ends of the visible spectrum. Such

results are also in stark contrast to a 2021 review [6] specifically of RL for acne vulgaris, which found no difference in treatment outcomes between treated and control groups.

Many of the biological mechanisms postulated for the ostensible efficacy of LED therapy remain highly speculative and poorly demonstrated, in stark contrast to treatments like ultraviolet therapy [7] or even ionizing radiation [8] where mechanisms of action are well-understood and not contested. This is compounded by significant inconsistencies in the reporting of what actual doses patients receive in LED therapy, and vast differences in wavelengths, fluences, and regimens that make direct comparison of treatments difficult. Accordingly, it becomes difficult to compare seemingly similar therapies, and even difficult to contrast them with the equivalent solar exposure required for the same fluence. To date, there has been little work done on the aspect of quantification of dose, despite the fact that this is a vital aspect to consider when ascertaining whether observed treatments effects are comparable. Even more crucially, such quantification is vital to ensure reported results are not simply spurious findings that do not benefit patients, especially when purposed mechanisms of action remain contested.

Accordingly, this work establishes a formal way of comparing patient received dose and applies this to previous studies to ascertain potential inhomogeneity in reported doses, establishing a framework to compare all treatment doses to the equivalent solar exposure time needed to achieve fluences reported at specified wavelengths in the red light (RL), blue light (BL), and yellow light (YL) portion of the spectrum. It also derives a means to compare this to typical solar irradiance at the Earth's surface to allow clear contextualisation of findings. Importantly, this is not intended to be a systematic review of these studies, and instead uses them as a convenience sample to illustrate the inherent difficult and nuance of the problem. This work confines itself to studies directly employing LEDs for treatment, rather than any intense pulsed light or laser therapies. Equally, photodynamic therapies (PDT) involving the activation of a chemical agent are not considered in this work, though some of the dosimetric considerations discussed here may apply.

## Materials and methods

### Determination of spectral widths and derivation of solar equivalent dose

RL, YL and BL LED devices are not monochromatic, and are typically specified with a central wavelength, $\lambda_c$. LEDs have a spectral width given by the Full Width at Half Maximum (FWHM), which for a typically normally distributed (or approximately normally distributed) LED profile is related to standard deviation by FWHM $\approx 2.355\sigma$. Accordingly, for any central wavelength $\lambda_c$, the proportion of light energy within any interval centred on this wavelength is a function of the Gaussian spectral width, as shown in Fig 1(a). This means that only a proportion of the quoted fluence will be emitted within any interval centred on a given wavelength, and it is important to account for this. For all included studies, wavelength and FWHM values were extracted when reported or available, and calculated from typical FWHM data for comparable modern LED sources [9] when not. For an LED source centred at $\lambda_c$ with total irradiance $E_T$ and fluence $D_T$, fluence also obeys a Gaussian distribution, and accordingly total fluence between any lower wavelengths ($\lambda_l$) and higher wavelength $\lambda_u$ in the band is akin to a Gaussian cumulative density function, given by

$$D(\lambda_l, \lambda_u) = \frac{D_T}{2}\left(\text{erf}\left(\frac{\lambda_u - \lambda_c}{\sqrt{2}\sigma_c}\right) - \text{erf}\left(\frac{\lambda_l - \lambda_c}{\sqrt{2}\sigma_c}\right)\right).$$

The American Society for Testing and Materials (ASTM) G-173 spectra for terrestrial solar spectral irradiance mimics the conditions and tilt angle of the average latitude for the contiguous USA, with the receiving surface defined as an inclined plane at 37° tilt toward the equator, facing the sun, with atmospheric conditions as defined in the standard and an absolute air mass of 1.5 (solar zenith angle 48.19°s). Spectral irradiance, the irradiance per wavelength, defined as $s(\lambda)$, has an integrated irradiance across the entire range of $1000.W/m^2$ a portion of which is shown in Fig 1(b) including diffuse sky radiation. To compare the ostensible therapeutic dose from a given RL/BL/YL source, reported irradiances and doses for

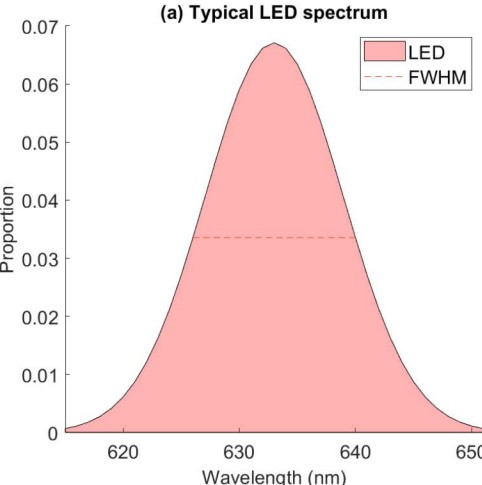
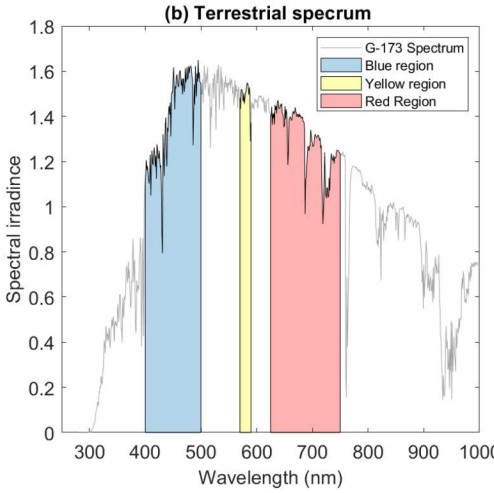

**Fig 1. (a)** Spectrum for a red LED with $\lambda_c = 633$ nm and FWHM = 14 nm, **(b)** Terrestrial solar spectrum with blue band (400–500 nm), yellow-band (570–590 nm) and red band (625–750 nm) highlighted.

each study were extracted, as was central wavelength and FWHM for the LEDs used. Values for $D$ were calculated for each 1 nm wavelength step along $\lambda \pm 3\sigma$, accounting for over 97% of the total LED output. For each step, the integrated solar irradiance from the G-173 spectra were also calculated with trapezoidal integration methods, given by

$$E_S(\lambda_u, \lambda_l) = \int_{\lambda_l}^{\lambda_u} S(\lambda) d\lambda.$$

Because LEDs are not monochromatic, it is important to calculate the given output at a quoted central wavelength, and find a suitable basis for comparison. In this work, we propose the equivalent solar exposure time, defined as the period of solar exposure required to achieve the same fluence in the same spectral band from an LED source, given by

$$t_S = \frac{D(\lambda_u, \lambda_l)}{E_S(\lambda_u, \lambda_l)}.$$

With this performed at 1 nm steps between $\lambda_c \pm 3\sigma$, the greatest value in the resultant vector (typically at central wavelength, λc) corresponded to the maximum solar exposure time required to achieve the same irradiance as a given LED treatment, facilitating direct comparison. Sample code to perform these calculations is available online at https://github.com/drg85/LEDcheck.

## Analysis of LED studies

The framework here was applied to studies of LEDs included in recent systematic reviews [5,6] which considered only LED therapies were assessed for information pertaining to condition treated, wavelength/s used, exposure time $t_L$) and device details. A further relevant study not included in these works due to later publication (2023–2024) was identified on Pubmed and included here. Total fluence in Joules per square centimetre per treatment, $D$, was extracted when directly reported, while for studies reporting irradiance ($E_L$, Watts per square centimetre), fluence was calculated by $D = E_L t_L$. When all quantities were reported, units were checked for consistency. Information regarding number of patients treated

and whether the study or its investigators were funded by light-treatment device manufacturers was also extracted. Papers were inspected to investigate whether they independently measured fluence or irradiance of the equipment used.

### Effective fluence for various full-width half maxima

A major potential methodological pitfall in LED PBM research is the implicit assumption that quoted fluences in literature correspond to actual fluence at the central wavelength. This is not true for reasons outlined above, and in review we additionally calculate actual effective fluences for LED sources with various FWHM to demonstrate the degree of error that might arise when this assumption is wrongly held.

### Results

In the 27 LED only studies analysed, 9 (33.3%) were on acne vulgaris, 5 (18.5%) on wrinkle reduction, 3 (11.1%) on wound healing, 3 (11.1%) on psoriasis severity, and 7 (25.9%) on erythema index rate. Total number of patients ranged from 14 to 105, with a median of 26 patients. Outcomes measures between these studies were highly heterogeneous and typically not amenable to direct comparison or synthesis. Of these studies, Fluences and wavelengths used in treatment differed starkly across studies, ranging from $0.1J\ cm^{-2} - 126J\ cm^{-2}$, a difference of over three-orders of magnitude. Similarly, the calculated equivalent solar time differed vastly, from 0.01–19.35 hours. Central wavelengths ranged from 405nm (BL) – 660nm (RL). In 33.3% ($n = 9$) of studies, assessors of outcome measures were unblinded and control / treatment group known, potentially introducing experimental bias. A further 33.3% ($n = 9$) did not have any non-light control group (NNLC) for comparison, instead comparing different wavelengths of light or light modalities to each other rather than standard treatment. The remaining studies (33.3%, $n = 9$) have both suitable control group and blinded outcome assessment. Histograms of the fluence and solar exposure time are shown in Fig 2, and study details are given in Table 1. None of the studies (100%, $n = 27$) reported any dose or fluence validation nor technical information on the LED specifications like FWHM, and 10 of the studies (37.0%) were sponsored by the device manufacturer, with a further 3 (11.1%) conducted by commercial dermatology practices offering the therapy under investigation as a treatment option. Table 2 shows the error in fluence assumptions for various FWHMs for varying typical LED sources.

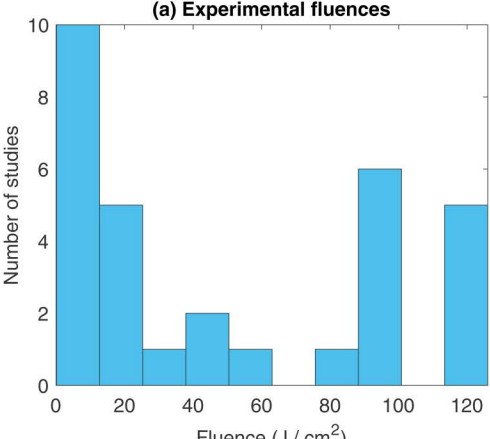 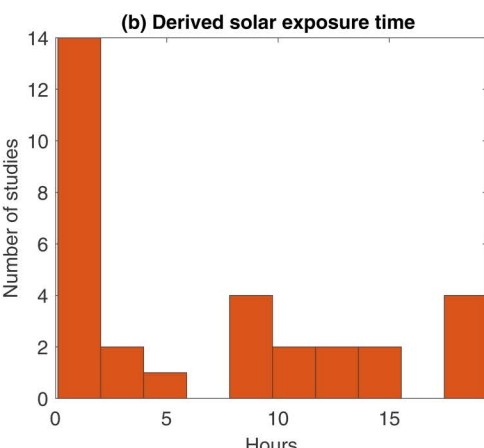

**Fig 2. (a) Histogram of reported fluences for included studios (b) Histogram of derived equivalent solar exposure time for included studies where data was available.**

**Table 1. Properties of LED PBM studies analysed.**

| Source | N | Manufacturer Sponsored | Assessor Blinded?[Θ] | Device[*] | $\lambda_c$ (FWMH) | Fluence D | Equivalent solar exposure (hrs) |
|---|---|---|---|---|---|---|---|
| **Acne Vulgaris Studies** | | | | | | | |
| Alba 2017 [10] | 22 | No | No | Spectra G3 | RL: 660 (20) BL: 470 (20) | RL: 8.0 $J\,cm^{-2}$ BL: 8.0 $J\,cm^{-2}$ | RL: 0.8 hrs BL: 0.7 hrs |
| Nestor 2016 [11] | 105 | Yes | No | Illumask La Lumiere | RL: 630 (14) BL: 445 (20) | RL: 17.9 $J\,cm^{-2}$ BL: 10.4 $J\,cm^{-2}$ | RL: 2.4 hrs BL: 1.0 hrs |
| Ash 2015 [12] | 41 | Yes | No | Dezac | BL: 414 (20) | BL: 17.6 $J\,cm^{-2}$ | BL: 1.1 hrs |
| Liu 2014 [13] | 50 | No | NNLC | Omnilux revive/blue | RL: 633 (12) BL: 414 (10) | RL: 126.0 $J\,cm^{-2}$ BL: 48.0 $J\,cm^{-2}$ | RL: 19.4 hrs BL: 10.3 hrs |
| Kwon 2013 [14] | 35 | No | Yes | OCimple MP 200 | RL: 620 (14) BL: 420 (20) | RL: 1.2 $J\,cm^{-2}$ BL: 0.9 $J\,cm^{-2}$ | RL: 0.2 hrs BL: 0.1 hrs |
| Gold 2011 [15] | 30 | Yes | No | TandaZap | BL: 414 (15) | Inadequate data[Δ] | Undetermined |
| Liu 2011 [16] | 20 | No | NNLC | Rainbow communications | RL: 630 (20) BL: 405 (20) | RL: 11.5 $J\,cm^{-2}$ BL: 7.2 $J\,cm^{-2}$ | RL: 1.1 hrs BL: 0.8 hrs |
| De Arruda 2009 [17] | 60 | No | No | Soret Blue light | BL: 410 (20) | BL: 36.0 $J\,cm^{-2}$ | BL: 4.2 hrs |
| Im 2007 [18] | 28 | No | Yes | SoftLaser SL30 | RL: 653 (35) | RL: 5.4 $J\,cm^{-2}$ | RL: 0.3 hrs |
| **Wrinkle reduction studies** | | | | | | | |
| Couturaud 2023 [19] | 20 | Yes | NNLC | Skin Light Dior | RL: 633 (20) | RL: 15.6 $J\,cm^{-2}$ | RL: 1.4 hrs |
| Nikolis 2016 [20] | 32 | Yes | NNLC | KLOX LED Light | BL: 446 (50) | BL: 45.0 $J\,cm^{-2}$ | BL: 2.2 hrs |
| Migliardi 2009 [21] | 30 | No | NNLC | LightActive Bimedica | RL: 633 (14) | Inadequate data[Δ] | Undetermined |
| Lee 2007 [22] | 76 | No[o] | Yes | Omnilux Plus | RL: 633 (12) | RL: 126.0 $J\,cm^{-2}$ | RL: 19.4 hrs |
| Bhat 2005 [23] | 23 | Yes | Yes | Omnilux Revive | RL: 633 (12) | RL: 96.0 $J\,cm^{-2}$ | RL: 14.7 hrs |
| **Wound-healing studies** | | | | | | | |
| Perper 2020 [24] | 14 | No | Yes | Omnilux model | RL: 633 (12) | RL: 126.0 $J\,cm^{-2}$ | RL: 19.4 hrs |
| Siqueira 2015 [25] | 17 | No | Yes | Custom device | RL: 625 (5) | RL: 4.0 $J\,cm^{-2}$ | RL: 1.4 hrs |
| Lei 2015 [26] | 26 | No | No | Omnilux Red | RL: 633 (12) | RL: 80.0 $J\,cm^{-2}$ | RL: 12.3 hrs |
| **Psoriasis Severity studies** | | | | | | | |
| Pfaff 2015 [27] | 47 | Yes | NNLC | Phillips device | BL: 453 (20) | BL: 90.0 $J\,cm^{-2}$ | BL: 7.9 hrs |
| Kleinpenning 2012 [28] | 20 | Yes | NNLC | Phillips devices | RL: 630 (14) BL: 414 (10) | RL: 60.0 $J\,cm^{-2}$ BL: 120.0 $J\,cm^{-2}$ | RL: 7.9 hrs BL: 13.3 hrs |
| Weinstable 2011 [29] | 20 | Yes | NNLC | Phillips devices | BL: 453 (20) BL: 420 (20) | RL: 90.0 $J\,cm^{-2}$ BL: 90.0 $J\,cm^{-2}$ | RL: 7.9 hrs BL: 10.0 hrs |
| **Erythema index rate studies** | | | | | | | |
| Wanit. 2019 [30] | 19 | No | Yes | Omnilux model | RL: 633 (12) | RL: 126.0 $J\,cm^{-2}$ | RL: 19.4 hrs |
| Keemss 2016 [31] | 21 | Yes | No | Phillips Device | BL: 453 (20) | BL: 90.0 $J\,cm^{-2}$ | BL: 7.9 hrs |
| Alster 2009 [32] | 20 | No[ɫ] | Yes | Gentlewaves | YL: 590 (20) | YL: 0.1 $J\,cm^{-2}$ | YL: <0.1 hrs |
| Khoury 2008 [33] | 15 | No[ɫ] | Yes | Gentlewaves | YL: 590 (20) | YL: 17.0 $J\,cm^{-2}$ | YL: 1.5 hrs |
| Sasaki 2007 [34] | 15 | No | NNLC | Unspecified LED | RL: 660 (20) | Inadequate data[Δ] | Undetermined |
| Deland 2007 [35] | 19 | No[ɫ] | No | Gentlewaves | YL: 590 (20) | YL: 15.0 $J\,cm^{-2}$ | YL: 1.5 hrs |
| Trelles 2006 [36] | 28 | No | No | Omnilux model | RL: 633 (12) | RL: 96.0 $J\,cm^{-2}$ | RL: 14.7 hrs |

Θ Assessor blinded status – NNLC denotes No Non-Light Control, typically comparison of wavelengths with no other control.

*Device details/ manufacturers data was often incomplete, some devices obsolete or specifications No longer available.

Δ Inadequate fluence, irradiance, and/ or time of exposure data reported in study, solar equivalent indeterminable.

o No financial sponsorship, device supplied by company for purposes of study.

ɫ Research from dermatological practices offering the service.

**Table 2. Typical LED fluence deviation from quoted central wavelength.**

| FWHM | Fraction of Quoted Fluence at Central Wavelength |
|---|---|
| $FWHM = 5nm$ | 18.8% |
| $FWHM = 10nm$ | 9.4% |
| $FWHM = 12nm$ | 7.8% |
| $FWHM = 14nm$ | 6.7% |
| $FWHM = 20nm$ | 4.7% |
| $FWHM = 35nm$ | 2.7% |
| $FWHM = 50nm$ | 1.9% |

## Discussion

This work outlines a useful metric for the direction comparison of dose received from LED sources in the visible spectrum to facilitate synthesis of knowledge. Equally, it shows that there is vast methodological inconsistency in experiments to date, a fundamental issue to be addressed before any deeper understanding can be garnered. In the literature analysed, there was scant justification for many of the fluences and even wavelengths used. In the former case, quoted or derived fluence rates varied by over three orders of magnitude, even for treatments ostensibly for the same condition. This was not justified in the texts, and raises serious questions over the biological rationale behind these choices. The same issue exists with respect to wavelength; several of the studies reported efficacious results for both RL and BL. But these are at opposite ends of the visible spectrum, and there was scant biological discussion as to why both might be effective, nor discussion of the relatively small samples involved. The heterogeneity of outcomes analysed makes synthesis different, and 66.6% of the reported studies either had unblinded assessment that could potentially introduce bias, or did not compare the LED source to standard treatment.

This equivalent solar exposure metric deployed in this work has additional utility beyond quantification of different experiments. As all the wavelengths concerned are present in the terrestrial solar spectrum, then it is important to quantify how these apparently therapeutic doses compare to normal solar exposures. In some instances, the effective exposure of a treatment was much less than the equivalent solar exposure a patient would get from under an hour of normal diffuse natural light. If LEDs therapies do have a therapeutic effect, a failure to account for this would risk confounding all exposures in both control and active arms, when even relatively minuscule amounts of normal light would deliver much more fluence at those wavelengths in some instances. It raises a crucial biological question of what actual mechanism is thought to be behind apparent benefits quoted, and whether experiments were adequately designed to answer these questions.

This work also has limitations that need to be elucidated. Firstly, while the equivalent solar exposure time will have the same fluence and central wavelength as a given quasi-monochromatic LED, the non-Gaussian nature of solar spectra means the precise wavelength distribution will be a function of the solar irradiance spectra. The solar equivalent dose calculation also pivots on the fundamental assumption that the LED spectra is approximately Gaussian. This is however a reasonable assumption, and most LEDs are sold with their FWHM quoted on this basis [9]. Some LEDs have minor asymmetry in their output profile at certain wavelengths [37,38], and while this should not change estimations here, there may be situations where an LED source is for some reason non-Gaussian, in which case such calculations would be inaccurate. But equally, this raises further methodological questions about the studies considered, as few gave adequate information on source properties or reported validating the dose. It is worth noting that fluence estimates in all studies are inherently optimistic. All studies took the stated device power without clarifying whether that power referred to the device's electrical power or optical power output, with the latter presumed in all studies. If the former was instead the case, actual fluences would be markedly lower than quoted. None of the studies reported any independent fluence verification, however.

Another potential limitation is that the work outlined assumes all devices are centred around a particular wavelength in red, blue, or yellow, which is determined by the LED type. This does indeed appear to be the case, but it is also possible that a hypothetical device could use a white LED and then a colour filter, in which case the line shape will be influenced by the combination of the white LED emission and the filter transmission. This could be markedly different from the Gaussian profile assumptions elucidated here, as the emission from white LEDs is due to phosphors excited by the emission from a ultraviolet LED at the centre of the structure, a physical emission process entirely different to an electrically injected LED with emission determined solely by semiconductor bandgap. White LEDs are however considerably more expensive than quasi-monochromatic LEDs, as the requirement for a phosphor coating drives up prices, and also have much lower efficiency. For this reason, devices for which specifications were available simply used quasi-monochromatic LEDs for PMB. In these cases, the analysis presented here holds, but the paucity of reporting of the technical specifications of many of these devices can make this hard to determine.

It is also worth noting despite the limitations of many of these studies and the fact many are relatively old, from between 2005–2023, this has not stopped them being embraced by medical influencers and beauty bloggers online as evidence of efficacy. An abundance of influencers, some of whom have medical backgrounds, have endorsed expensive LED treatment devices claiming such studies show their efficacy, despite highly conflicting evidence. In 2024, red light therapy trended on TikTok with upwards of 70 million view [4], and devices being offered ranging from $100 to $3500. Some of this enthusiasm is motivated by lucrative sponsorship deals, but it is most certainly exacerbated by weak studies being over interpreted without their limitations being either understood or elucidated, such as the weaknesses outlined here.

While PBM is certainly worth exploring and may have medical application, inadequately reported or poorly conducted studies contribute to research waste [39], make errors harder to correct [40], and ultimately confound the public. In biomedical science, there is dawning awareness of much literature being non-replicable, with detrimental consequences for all. Nor can meta-analysis and systematic reviews undo poor quality studies when the reporting is wildly inconsistent, as it becomes extremely difficult to compare effects. This problem is at the heart of this work, and should serve as a case study for why it is critical to compare cautiously, with many systematic reviews greatly overestimating effect size [41,42] and themselves becoming arguably research waste.

## Conclusions

This framework here allows cross comparison of existing LED studies, and provides strong evidence that current undertakings are highly heterogeneous and potentially wasteful. Future endeavours should be cautiously reported in a consistent manner, and a biological rationale for particular fluences, wavelengths, and trials should be clearly elucidated, lest researchers end up misleading themselves and the public with potentially spurious findings.

## Acknowledgments

DRG would like to thank Prof Enda McGlynn, School of Physical Sciences, Dublin City University, for his helpful discussions on LED light profiles.

## Author contributions

**Conceptualization:** David Robert Grimes.

**Formal analysis:** David Robert Grimes.

**Funding acquisition:** David Robert Grimes.

**Methodology:** David Robert Grimes.

**Software:** David Robert Grimes.

**Writing – original draft:** David Robert Grimes.

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
