## [Decision Letter · Decision Letter 0]

20 Aug 2025

Dear Dr. Grimes,

Thank you for submitting your manuscript to PLOS ONE. After careful consideration, we feel that it has merit but does not fully meet PLOS ONE’s publication criteria as it currently stands. Therefore, we invite you to submit a revised version of the manuscript that addresses the points raised during the review process.

We look forward to receiving your revised manuscript.

Kind regards,

Taher Hatahet, Ph.D

Academic Editor

PLOS ONE

Journal Requirements:

[This work has been funded by the Wellcome Trust (Grant number 214461/A/18/Z).]. 

3. Thank you for stating the following in your manuscript:

[This work has been funded by the Wellcome Trust (Grant number 214461/A/18/Z).]

[This work has been funded by the Wellcome Trust (Grant number 214461/A/18/Z).]

4. In the online submission form, you indicated that [Code to run all the analysis in this work is hosted on GitHub ( https://github.com/drg85/LEDcheck ) and further analytical software in various languages is available by request from the author.].

Reviewers' comments:

Reviewer's Responses to Questions

**Comments to the Author**

1. Is the manuscript technically sound, and do the data support the conclusions?

Reviewer #1: Yes

2. Has the statistical analysis been performed appropriately and rigorously?

Reviewer #1: Yes

3. Have the authors made all data underlying the findings in their manuscript fully available?

Reviewer #1: Yes

4. Is the manuscript presented in an intelligible fashion and written in standard English?

Reviewer #1: Yes

Reviewer #1: It is interesting paper and important especially at academic as well as social point of view. As the author mentions, standardization of light characteristic is mandatory to evaluate impacts of therapeutic modalities with light (LED of course or even UV light) on exposed human body. Without knowing this, we, not only non-professional uses but also medical professional, might just feel like we understand it. Particular interest should be focused on comparison with hours of equivalent solar exposure time. This may objectify fluence of each LED tools. Only question to the author is difference of non-Gaussian- from Gaussian-LED light spectrum when compared with equivalent solar exposure. I would think that comparison with non-Gaussian light spectrum might be much difficult and complicated than Gaussian ones. Of course, as a matter of fact, I wouldn't think that non-Gaussian LED spectrum is superior, however, it is better stated in the discussion section when such devices are available in the market presently.

**Do you want your identity to be public for this peer review?** For information about this choice, including consent withdrawal, please see our Privacy Policy

Reviewer #1: No

---

## [Editor Report · Decision Letter 1]

8 Sep 2025

Methodological issues in visible LED therapy dermatological research and reporting

PONE-D-25-29256R1

Dear Dr. Grimes,

We’re pleased to inform you that your manuscript has been judged scientifically suitable for publication and will be formally accepted for publication once it meets all outstanding technical requirements.

Kind regards,

Taher Hatahet, Ph.D

Academic Editor

PLOS ONE

Additional Editor Comments (optional):

Thanks for working thorugh the comments
---

## [Editor Report · Acceptance letter]

PONE-D-25-29256R1

PLOS ONE

Dear Dr. Grimes,

I'm pleased to inform you that your manuscript has been deemed suitable for publication in PLOS ONE. Congratulations! Your manuscript is now being handed over to our production team.

Kind regards,

on behalf of

Dr. Taher Hatahet

Academic Editor

PLOS ONE